# Electrochemical Determination of Morphine in Urine Samples by Tailoring FeWO_4_/CPE Sensor

**DOI:** 10.3390/bios12110932

**Published:** 2022-10-27

**Authors:** Miloš Ognjanović, Katarina Nikolić, Marko Bošković, Ferenc Pastor, Nina Popov, Marijan Marciuš, Stjepko Krehula, Bratislav Antić, Dalibor M. Stanković

**Affiliations:** 1VINČA Institute of Nuclear Sciences-National Institute of the Republic of Serbia, University of Belgrade, Mike Petrovića Alasa 12-14, 11000 Belgrade, Serbia; 2Faculty of Chemistry, University of Belgrade, Studentski trg 12-16, 11000 Belgrade, Serbia; 3Division of Materials Chemistry, Ruđer Bošković Institute, 10000 Zagreb, Croatia

**Keywords:** iron tungstate, carbon paste electrode, electroanalysis, square wave voltammetry, morphine, real-world sample

## Abstract

Morphine (MORPH) is natural alkaloid and mainly used as a pain reliever. Its monitoring in human body fluids is crucial for modern medicine. In this paper, we have developed an electrochemical sensor for submicromolar detection of MORPH. The sensor is based on modified carbon paste electrode (CPE) by investigating the Fe_x_W_1-x_O_4_ ratio in iron tungstate (FeWO_4_), as well as the ratio of this material in CPE. For the first time, the effect of the iron–tungsten ratio in terms of achieving the best possible electrochemical characteristics for the detection of an important molecule for humans was examined. Morphological and electrochemical characteristics of materials were studied. The best results were obtained using Fe_1_W_3_ and 7.5% of modifier in CPE. For MORPH detection, square wave voltammetry (SWV) was optimized. Under the optimized conditions, Fe_1_W_3_@CPE resulted in limit of detection (LOD) of the method of 0.58 µM and limit of quantification (LOQ) of 1.94 µM. The linear operating range between 5 and 85 µM of MORPH in the Britton–Robinson buffer solution (BRBS) at pH 8 as supporting electrolyte was obtained. The Fe_1_W_3_@CPE sensor resulted in good selectivity and excellent repeatability with relative standard deviation (RSD) and was applied in real-world samples of human urine. Application for direct MORPH detection, without tedious sample pretreatment procedures, suggests that developed electrochemical sensor has appeared to be a suitable competitor for efficient, precise, and accurate monitoring of the MORPH in biological fluids.

## 1. Introduction

Opium poppy (lat. *Papaver somniferum* L.) is a species from the family *Papaveraceae*, native to eastern Mediterranean, but it is naturalized in Europe and Asia. The name “poppy” itself means “sleep” (lat. *Somnum*). It is predominantly a wild plant, widespread throughout the world, but is most prevalent in temperate and subtropical parts of the world. Opium poppy is a very hardy plant, which is easy to grow, and two main products are obtained from it: seeds and opium. *Papaver somniferum* seeds are harmless and are used as a spice during cooking, whereas opium contains a wide range of alkaloids and can be addictive. During the poppy harvest, the cocoon is cut and a brown sticky substance is released, which is opium glue. It is throbbed and homogenized into blocks, and the product thus obtained is crude opium, from which MORPH, thebaine, codeine, and papaverine are further extracted and used in clinical treatment. The use of opium poppy dates back to the ancient Minoans. Evidence of early domestication of this plant was discovered through small botanical remains found in the Mediterranean area and dates back to 5000 BC. It has been used to treat asthma, stomach ailments, and poor eyesight. MORPH as the basic alkaloid of opium [1] is used to relieve very severe pain after surgery [2], as well as in moderate to severe chronic pain in cancer patients [3]. MORPH is a potent opioid painkiller that affects the central and peripheral nervous systems by acting on opioid receptors [4]. Treatment with opioid drugs is complicated by their side effects. MORPH, when incorrectly dosed, can be toxic and addictive, which is why narcotic drugs are often abused. For this reason, this drug is on the black list of doping drugs in athletes [1]. That is why it is necessary to develop a method that easily and quickly determines the concentration of MORPH in blood and urine samples [5,6,7].

The methods that are usually used for the detection and quantification of MORPH are high-preformation liquid chromatography (HPLC), gas chromatography (GAC), surface plasmon resonance (SPR), radioimmunoassay (RAI), immunoassays, capillary electrophoresis (CE), and electrochemical methods. The most used are chromatography methods, but they need expensive equipment, reagents, and long analysis time.

The electrochemical method gives us a possibility of real-time and on-site analysis, where equipment is not expensive and is easy to handle [8,9,10]. In addition, with the development of nanomaterials, we obtain the ability to synthesize a large number of high-performance electrochemical sensors in order to create the most sensitive and selective sensors for a specific analyte [8,11,12,13,14,15]. Over the years, many electrochemical sensors based on glassy carbon electrodes [8,10,13,14,16], platinum electrodes, and carbon-based electrodes have been developed [9,10,11,12,17,18,19]. As modifiers for the electrochemical detection of MORPH, some of the following modifiers were used: cobalt–hexacyanoferrate particles, indium tin oxide modified with Prussian dye, carbon nanotubes and chitosan, palladium aluminum, multiwall nanotubes, and graphene dope [5,6,7,20,21].

In this paper, we constructed a CPE modified with FeWO_4_ and developed a method for the detection of MORPH under biological condition. Materials with different Fe/W ratios were synthesized, and their electrochemical and morphological properties were investigated. After the selection of an appropriate modifier, working conditions were optimized. Square wave voltammetric method was optimized and successfully applied for the detection of MORPH in urine samples, with satisfactory accuracy and precision and negligible matrix effect.

## 2. Materials and Methods

### 2.1. Reagents and Apparatus

Ammonium sulfate hexahydrate ((NH_4_)_2_Fe(SO_4_)_2_·6H_2_O, ACS reagent, 99%) and sodium tungstate dihydrate (Na_2_WO_4_·2H_2_O, ACS reagent, ≥99%) were purchased from Sigma Aldrich (St. Louis, MO, USA) and used as supplied without any further manipulation.

Standard MORPH solution (Sigma Aldrich, St. Louis, MO, USA) of 1∙10^−3^ M is prepared in a volumetric flask by taking a proper amount of MORPH and dissolved with distillated water. During electrochemical measurements, 3 mL of standard MORPH solution is used with 27 mL of a supporting electrolyte. As the supporting electrolyte, BRBS is used, which was prepared by mixing 0.4 M boric acids, phosphoric acids, and acetic acids. The values of pH for this buffer were adjusted with 0.5 M solution of sodium hydroxide.

The crystal structure of FeWO_4_ was investigated using X-ray powder diffraction (XRD) data collected on a high-resolution SmartLab^®^ X-ray diffractometer (Rigaku, Tokyo, Japan) operated at 30 mA current and 40 kV voltage. For the measurements, CuKα radiation source was used. Dried powders were placed on a silicon plate with zero background. Diffraction data were collected within 10–67° 2θ with a recording speed of 1°/min and step of 0.02°. The average crystallite size was estimated by applying Scherrer’s equation on the most intensive diffraction peaks [22]. The material morphology and energy-dispersive X-ray spectroscopy (EDS) elemental analysis used for electrode modification were investigated on a 10 keV field-emission–scanning electron microscope (FE-SEM) JSM 700F (JEOL, Tokyo, Japan). The samples were fixated on a holder with a conductive, followed by vacuum-drying and spray-coating with gold using a Sputter coater (Thermo-Fisher Scientific, Waltham, MA, USA). All electrochemical measurements were performed on Metrohm Autolab (Metrohm Autolab BV, Utrecht, The Netherlands) in a system with three electrodes, where working electrode was modified CPE/CPE, reference electrode Ag/AgCl (3 M KCl), and supporting Pt wire.

### 2.2. Synthesis of FeWO_4_

The synthesis of FeWO_4_ was performed following the work of Wang et al. with slight modifications [23]. Briefly, Fe*_x_*W_1-*x*_O_4_ mixed oxides were synthesized via a hydrothermal method in a 100 mL stainless-steel autoclave with a Teflon liner. Different Fe/W molar ratios (1:1, 1:3, and 3:1) were dissolved in 50 mL Milli-Q water. The mixture was sonicated for 30 min, and the pH of the solution was adjusted to 6 by adding either 1 M NaOH or H_2_SO_4_ solution. The autoclaves were heated at 180 °C for 5 h and cooled down to room temperature (RT). At the end, sediments were centrifuged and washed several times with Milli-Q water, and part of the samples was dried at 80 °C overnight for morphological and microstructure analysis.

### 2.3. Carbon Paste Electrode Preparation

For the preparation of unmodified CPE, 40 mg of carbon powder (Sigma Aldrich, St. Louis, MI, USA) and 10 µL of paraffin oil were used. The ingredients were mixed until a paste was formed and left to stand overnight at room temperature. After that, a part of the composite mixture was packed at the end of the Teflon electrode body (d = 3 mm).

Modified electrodes were prepared by measuring carbon powder and modificatory FeWO_4_ in different mass ratios (2.5%, 5%, 7.5%, and 10%) so that the combined mass is 40 mg. To the mixture, 10 µL paraffin oil was added. All ingredients were mixed together until a homogeneous paste was formed and left on room temperature overnight. During the process, a part of the composite mixture was densely packed at the end of Teflon electrode body.

## 3. Results and Discussion

### 3.1. Physicochemical Characterization

The crystal and phase structure of the synthesized materials were analyzed by XRD (Figure 1). The sample Fe_1_W_3_ (blue line) crystallizes in the pure wolframite-like monoclinic crystal structure of FeWO_4_ (JCPDS file no. 74-1100) without visible impurities of tungsten oxide and iron oxide [23]. The average crystallite size of Fe_1_W_3_ was 19.7 nm, estimated by the Scherrer equation. Similarly, the Fe_1_W_1_ sample (green line) also crystallizes in the monoclinic crystal structure of FeWO_4_. Compared with Fe_1_W_3_, the crystallinity of this material is much lower given the broad diffraction pattern, and the crystallite size was substantially smaller at 7.3 nm. On the other hand, sample Fe_3_W_1_ (red line) crystallizes in two phases; apart from FeWO_4_, hexagonal α-Fe_2_O_3_ (JCPDS file no. #86-055) is also present [24]. The average size of the crystallite is 25.7 nm. Given all said so far, Fe_1_W_3_ is the purest iron tungstate of the prepared, and it is expected to serve as the best electrocatalyst of all samples.

The morphology study of the prepared materials has been displayed in Figure 2. As can be seen, there are distinctive differences as a consequence of varying Fe/W molar ratios (1:1, 1:3, and 3:1) in FeWO_4_ during the synthesis process. Fe_3_W_1_ nanoparticles are in the form of nanorods a few hundred nanometers long and ~50 nm wide, which are specifically aggregated into ball-like structures about 500 nm in size (Figure 2a). The Fe_1_W_1_ nanoparticles have no distinctive rod-like form, but similarly to Fe_3_W_1_, they aggregate into spherical agglomerates (Figure 2b). On the other side, Fe_1_W_3_ nanoparticles are considerably different, with two distinctive morphologies, well-dispersed and without observable assembly of nanoparticles (Figure 2c). One portion of Fe_1_W_3_ is formed of ~50 nm wide and ~400 nm long needles, while the rest is composed of pill-shaped nanoparticles about 50 nm in size. With an increase in the share of tungsten in the composite material, the narrowing and at the same time an elongation of the nanorods are evident, which increase the specific surface area of the material. In addition, a reduction in their assembly into spherical structures can be observed. Figure 2D–F reveal the elemental analysis of the samples, with a Fe/W ratio of 0.83:0.17, 0.55:0.45, and 0.45:0.55 in Fe_3_W_1_, Fe_1_W_1_, and Fe_3_W_3_, respectively. This is in agreement with the molar ratios set during the synthesis of nanoparticles.

### 3.2. Electrochemical Characterization

The electrochemical properties of the modified electrodes and the effect of the type of the modifier were investigated by utilizing cyclic voltammetry and electrochemical impedance spectroscopy (EIS) in the 5 mM Fe^2+/3+^ in 0.1 M potassium chloride as the supporting electrolyte. For modificators, FeWO_4_ was used, where the ratios of Fe and W are varied; modifier 1 (Fe_1_W_1_) has a ratio Fe/W = 1/1, modifier 2 (Fe_1_W_3_) Fe/W = 1/3, and modifier 3 (Fe_3_W_1_) Fe/W = 3/1. As can be seen in Figure 3A, an increase in both anode and cathode currents was obtained using an electrode modified with Fe_1_W_3_ material compared with unmodified CPE, Fe_1_W_1_/CPE, and Fe_3_W_1_/CPE. Using this electrode, well-defined reversible peaks were obtained, with clearly noticeable increase in the peak current and significant decrease in the peak-to-peak potential. Higher presence of tungsten in the material probably improves the electronic environment at the electrode surface and enables fast electron transfer, increased conductivity, and excellent electrocatalytic behavior. To confirm this observation, we employed EIS in the same testing solution. Results are given in Figure 3B. The modification of the CPE with prepared materials would evidence the decrement in charge-transfer resistance and diffusion resistance at the electrode–electrolyte interface (bare 17 kΩ, F_1_W_1_ = 7.4 kΩ, F_1_W_3_ = 1.6 kΩ, and F_3_W_1_ = 8.5 kΩ). The lowest R_ct_ values for Fe_1_W_3_ electrode indicate that selected material strongly improved charge transport at the electrode–solution interface, increasing the active surface area and diffusion layer.

The electrochemical behavior of MORPH (0.1 mM) was examined by CV in BRBS with pH 8 on the surface of CPE modificated with three different modifications. Results of all three modifications are summarized in Figure 3C. As can be seen, in the anode direction, two well-defined and clearly visible oxidation peaks are noticed. First, a clearly defined peak (at potential around 0.4 V) comes from the transition of one electron during the oxidation of the phenolic group in position 3, which leads to the formation of pseudomorphine as the main product. The suggested oxidation mechanism for this peak is shown in Figure 1.

This peak is most intense and sharpest during measurements with Fe_1_W_3_/CPE, considering that this and low peak potential future measurements worked with this electrode. Besides that, at a potential around 0.85 V, we have another peak of oxidation that can be related with the oxidation of the second −OH group. In another direction, there is an absence of reduction peaks, which brings us to the conclusion that the oxidation of MORPH is an irreversible process. This behavior can be attributed to the FeWO_4_ (Fe_1_W_3_) at the electrode surface, which increases the electrode active surface area and promotes the diffusion capacities of the electrode, whereas bare CPE is made of a heterogeneous carbon material that very often exhibits slower electron transfer than homogeneous carbon electrodes. It can be concluded that Fe_1_W_3_ can serve as an effective electrode modifier, acting as a current promoter and surface conductivity enhancer, and Fe_1_W_3_/CPE was selected as the basis for further experiments.

After that, it was examined how the optimum percentage of a modifier can affect the determination of MORPH. Results are shown in Figure 3D. Optimizing a percentage of modifiers is an important factor when developing an electrochemical method, as it directly affects the amount of chemicals required for the method to successfully work—the greenness of the technique. The obtained results indicate that an increase in the percentage of modifiers from 2% to 10% leads to an increase in current. However, it can be concluded that 5% of the modifier gives satisfactory results and that, with this electrode, the additional peaks obtained during the oxidation of MORPH are better seen. Based on all of the above, the CPE with 7.5% of the material was chosen to optimize the analytical procedure for the MORPH detection.

It is well-known that the analytical properties of the electrochemical method are closely related to the structure and characteristic of the supporting electrolyte. To examine the impact and evaluate the response of Fe_1_W_3_/CPE on MORPH oxidation, BRBS buffer was used in range of 2 to 10. Results can be found in Figure 4A. MORPH was responding in the whole tested pH range. On pH values over 8, slight decreasing strength of the peak current was noticed (Figure 4B). That confirmed that the proton participates in the electrochemical reaction of MORPH oxidation and is also obtained by the dependence of the oxidation potential of the first peak in relation to pH. There is a linear dependence that can be represented by the equation *E_p_* = −0.061∙pH + 0.891. The obtained slope value of 61 mV is very close to the ideal theoretical value of 59 mV, suggesting that the same number of protons and electrons participates in the electrochemical reaction (Figure 4C). This is fully in accordance with the proposed mechanism for MORPH oxidation. Since the aim of the research was to implement the method in real biological samples, pH 8 was used in further studies as this value is close to the physiological pH.

In order to further investigate the properties of the modified electrode and to study the nature of the electrode reaction, the electrochemical oxidation of MORPH as a function of different scan rates was investigated. The results are presented in Figure 4D. A variation in the scan rate from 50 to 200 mV/s resulted in a linear relationship between the peak current and scan rate. This linearity indicates that the electrode reaction on the Fe_1_W_3_/CPE surface is controlled by adsorption (Figure 4E). In addition, this is confirmed with the slight shift in peak potential with increasing scan rate toward more positive values.

### 3.3. Optimization of Square Wave Voltammetry (SWV) Instrumental Parameters for MORPH Determination

The most used electrochemical methods are different pulsed voltammetry (DPV) and square wave voltammetry. Comparing those two methods for 0.1 mM MORPH detection, the SWV method obtained a higher current peak and that method was selected for additional optimization. With the aim to achieve the best analytical performance of the detection method, suitable parameters for the SWV method were studied in the BRBS solution at pH 8 containing 0.1 mM of MORPH. Variations of step pulse amplitude in the range from 10 to 100 mV and frequency from 10 to 100 mV were carried out. When one parameter was optimized, others were kept constant. It was found that the optimization of each parameter promoted an increase in the oxidation current up to a certain value, after which it starts to decrease, followed by the increase in the background current and usually widening in the peak. The compromise values were as follows: pulse amplitude of 50 mV and frequency 75 mV (Figure 5A,B). Thus, these parameters were used for the construction of the calibration curve and method development.

### 3.4. Analytical Parameters of the Detection Method

Parameter optimization was a necessary condition for the estimation of the analytical performance of the detection method. For this purpose, a certain amount of the MORPH standard solution was added in the electrochemical cell, filled with BRBS at pH 8, and SWV was recorded under optimized conditions. The SWV voltammograms are shown in Figure 6A, whereas the corresponding calibration curve was constructed and shown in Figure 6B. In the concentration range from 5 µM to 85 µM of MORPH, the calibration curve shows excellent linearity with the corresponding linear equation $I_{A}=2.025\cdot 10^{-8}c-1.1\cdot 10^{-7}$, where *I_A_* is the oxidation current, and *c* is the concentration of MORPH in µM. The regression coefficient for this linearity was *R^2^* = 0.993. The limit of detection (LOD) and limit of quantification (LOQ) were calculated as $\mathrm{LOD}=\frac{3\;\mathrm{SD}}{\mathrm{slope}}$ and $\mathrm{LOQ}=\frac{10\mathrm{SD}}{\mathrm{slope}}$, respectively. In these equations, *SD* is given as the standard deviation of the blank. The LOD of the method was calculated to be 0.58 µM, and the LOQ was 1.94 µM. When comparing our results with the results found in today’s literature in terms of detection limit and linear range (Table 1), it can be concluded that the results obtained in this work are comparable with or better than the results of electrochemical sensors reported so far.

The prepared sensor exhibits excellent repeatability with similar activity for 10 independent measurements of the same concentration of MORPH (25 µM) using the same electrode. The obtained RSD for this study was 2.12%. Similarly, the reproducibility of five independently prepared electrodes was verified with the measurements of the same concentration of MORPH, giving an RSD of 2.56%. The stability of the electrode was monitored during the period of one month. Measurements of 25 µM of MORPH were performed daily using the same electrode kept in the refrigerator at 4 °C. The current decrease of 4.86% during this period can be assigned as negligible. These studies indicate that Fe_1_W_3_/CPE had proved stability, reproducibility, and repeatability toward MORPH detection.

### 3.5. Selectivity of Method and Application in Real-World Samples Analysis

Selectivity of the method is a crucial parameter for the application in real-word sample analysis. To investigate this, a possibility for the detection of 50 µM of MORPH was investigated in the presence of ascorbic acid (AA), glucose (GLU), dopamine (DOP), citric acid (CA), and uric acid (UA) under previously optimized conditions. Other alkaloids that may be present in human fluids show their electrochemical behavior at potentials that do not match the oxidation potential of morphine. Often, in the literature, they are simultaneously determined with morphine. Such studies are shown for the determination of morphine and codeine by Wester et al. [2] and Bagheri et al. [25]; morphine and methadone by Habibi et al. [26]; monoacetylmorphine, morphine, and codeine by Gerostamoulos et al. [27]; and morphine and fentanyl by Nazari and Eshaghi [28]… These compounds were selected as most commonly found in the biological fluid samples. Selected organic compounds in a ratio of 1:1 have an impact on the determination of MORPH. AA, UA, and GLU do not have a significant effect on the oxidation peak of MORPH, while CA, and DOP, reduces and expands the oxidation peak. All voltammograms and peak current signals are summarized in Figure 7.

The investigative constancy of the modified electrode was tested using a developed method for the detection of MORPH in the urine samples. Samples were taken from apparently healthy volunteers and directly tested. The accuracy of the method was validated with the high-performance liquid chromatography analysis (Thermo Scientific UltiMate 3000 HPLC with a diode array detector) based on the method proposed by Fernandez et al. [29]. Results obtained from HPLC measurements at wavelengths 250 and 283 nm are given in Figure 8. The sample volume of 1 mL was diluted three times with the supporting electrolyte and analyzed. The dilution was chosen according to literature data, where it was stated that the concentration of morphine in urine is up to 26 µm, several times lower than in blood samples (up to 100 µM) [30], so that a sample that was diluted three times would be in the lower limit of the linear range of the method [31,32]. Primarily, urine samples with no added standard solution of MORPH had been tested and no current response was found. Then the analyses proceeded with the addition of distinct quantities of MORPH into the urine samples, which eventually showed a characteristic anodic peak. The experimental conditions were the same as those in which all the above experiments had been performed. Results are summarized in Table 2. The obtained results and recovery values in the range from 99% to 101%, and good agreement with the reference HPLC method, indicate that the proposed sensor can be used as an excellent platform for monitoring MORPH content in the biological fluids with no or negligible matrix effect. The proposed approach can serve as the excellent basis for the further miniaturization of the method and for potential transfer from laboratory application to commercial use.

## 4. Conclusions

In summary, we reported facile preparation of novel material based on the iron tungstate with different ratios of tungsten and iron. Microstructural and morphological properties of the obtained materials were investigated. It was found that the increase in the tungsten concentration strongly influences the electrochemical characteristics of the material. Successful modification of the CPE is followed with excellent electrocatalytic properties of the modified electrode toward the detection of MORPH in biological samples, with wide dynamic range, good selectivity, and low limit of detection. Furthermore, satisfactory recovery ranges and good accuracy of the proposed electrocatalyst validate its efficiency in detecting MORPH in real-world samples; while the preparation of the sample itself is very simple, it does not require any special treatment other than simply diluting the sample with a supporting electrolyte. This work can open new approaches for MORPH detection in medical practice and can serve as a basis for technology transfer from laboratory practice to commercial applications.

## Data Availability

Not applicable.

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
