# Peer review of "Electrochemical Determination of Morphine in Urine Samples by Tailoring FeWO4/CPE Sensor"

_biosensors, 2022, doi:10.3390/bios12110932_

Round 1

Reviewer 1 Report

COMMENTS FOR THE AUTHOR:

Manuscript entitled "Electrochemical determination of pain reliever morphine in biological matrices by tailoring FeWO4/CPE sensor" submitted by Miloš Ognjanović, Katarina Nikolić, Marko Bošković, Ferenc Pastor, Nina Popov, Marijan Marciuš, Stjepko Kerhula, Bratislav Antić, and Dalibor Stanković, can be accepted for publishing in the Biosensors, after minor revision. 

In this study, it was developed an electrochemical sensor for submicromolar detection of Morphine (MORPH). The sensor is based on modified carbon paste electrode (CPE) by investigating the FexW1-xO4 ratio in iron tungstate (FeWO4) and the best material were obtained (Fe1W3). The detail characterization of the material was done, and the square wave voltammetry (SWV) was optimized for MORPH detection. Also, the obtained sensor was applied in real world samples of human urine. The optimization process was carried out using response surface methodology (RSM) and central composite design (CCD). The manuscript presents interesting results, which are relatively well organized and systematized, but the novelty and practical applicability of this study should be highlighted more. In my opinion, this manuscript should be published in your Journal after minor revision.

Here is a list of my general comments:

·        The novelty and practical applicability of this study should be highlighted more.

·        Define abbreviations at first mention. Abbreviations should be defined at first mention and then through the text use only the abbreviations not the full name (RSD, EDS, MORPH…)

·        Newer references should be included in the introduction and the discussion part. From 21 references, only 11 are from last five years.

·        It would be interesting to compare the removal efficiency of morphine presented in this study with the results of other authors if it is applicable.

·        Specific comments:

o   Line 14, 15: In the sentence the ratio FexW1-xO4 ratio”, please remove first word ration.

o   Line 20: Please add full name for abbreviation “BRBS”.

o   Line 21: Please add full name for abbreviation “RSD”.

o   Line 39: In the abstract the authors introduced the abbreviation MORPH for morphine. Here is the first time that word morphine is used so please introduce the abbreviation MORPH here also. Or do not used it at all.

o   Line 91: Please add full name for abbreviation “EDS”.

o   Line 134: Please delete letter P from abbreviation “XRPD”.

o   Line 151: Please delete double word “with in”.

o   Line 153: The abbreviation FE-SEM is here for the first time. Please include it in the materials and methods in the description of the SEM (line 93).

o   Line 174: Please use abbreviation for the cyclic voltammetry - CV, because it is already introduced (157).

o   Line 175: Please use abbreviation for the carbon paste electrode – CPE, because it is already introduced (109).

o   Line 183, 188, 196, 232, 260, 264, 266, 289, 290, 317, 320, 321: Please use abbreviation for the morphine – MORPH or do not introduce the abbreviation at all. Please check the whole manuscript.

o   Line 239, 241: Please introduce once the abbreviation of square wave voltammetry (SWV) not twice.

o   Line 267, 268: Please give da abbreviation of limit of detection and limit of quantification: “Limit of detection (LOD) and limit of quantification (LOQ) were calculated…”

o   Line 273: Please add full name for abbreviation “RSD” when you first mentioned.

o   Line 316: Please use abbreviation for the carbon paste electrode – CPE.

Author Response

Reviewer 1 comments:

Manuscript entitled "Electrochemical determination of pain reliever morphine in biological matrices by tailoring FeWO4/CPE sensor" submitted by Miloš Ognjanović, Katarina Nikolić, Marko Bošković, Ferenc Pastor, Nina Popov, Marijan Marciuš, Stjepko Kerhula, Bratislav Antić, and Dalibor Stanković, can be accepted for publishing in the Biosensors, after minor revision. 

In this study, it was developed an electrochemical sensor for submicromolar detection of Morphine (MORPH). The sensor is based on modified carbon paste electrode (CPE) by investigating the FexW1-xO4 ratio in iron tungstate (FeWO4) and the best material were obtained (Fe1W3). The detail characterization of the material was done, and the square wave voltammetry (SWV) was optimized for MORPH detection. Also, the obtained sensor was applied in real world samples of human urine. The optimization process was carried out using response surface methodology (RSM) and central composite design (CCD). The manuscript presents interesting results, which are relatively well organized and systematized, but the novelty and practical applicability of this study should be highlighted more. In my opinion, this manuscript should be published in your Journal after minor revision.

We are very grateful for the positive and constructive comments.

Here is a list of my general comments:

  • The novelty and practical applicability of this study should be highlighted more.

Thank you for your suggestion. In the revised version we highlighted the novelty of the work and practical applicability, in the sections Abstract and Conclusions. All changes are marked in the revised version.

  • Define abbreviations at first mention. Abbreviations should be defined at first mention and then through the text use only the abbreviations not the full name (RSD, EDS, MORPH…)

Thank you for noticing this. In the revised version we defined abbreviations appropriately.

  • Newer references should be included in the introduction and the discussion part. From 21 references, only 11 are from last five years.

Thank you for the suggestion. Reference section is improved according the comments.

  • It would be interesting to compare the removal efficiency of morphine presented in this study with the results of other authors if it is applicable.

Thank you for your comment. In the revised version we provided table (the new Table 1 in this version) with comparison of our method with the current literature data.

  • Specific comments:

o   Line 14, 15: In the sentence “the ratio FexW1-xO4 ratio”, please remove first word ration.

o   Line 20: Please add full name for abbreviation “BRBS”.

o   Line 21: Please add full name for abbreviation “RSD”.

o   Line 39: In the abstract the authors introduced the abbreviation MORPH for morphine. Here is the first time that word morphine is used so please introduce the abbreviation MORPH here also. Or do not used it at all.

o   Line 91: Please add full name for abbreviation “EDS”.

o   Line 134: Please delete letter P from abbreviation “XRPD”.

o   Line 151: Please delete double word “with in”.

o   Line 153: The abbreviation FE-SEM is here for the first time. Please include it in the materials and methods in the description of the SEM (line 93).

o   Line 174: Please use abbreviation for the cyclic voltammetry - CV, because it is already introduced (157).

o   Line 175: Please use abbreviation for the carbon paste electrode – CPE, because it is already introduced (109).

o   Line 183, 188, 196, 232, 260, 264, 266, 289, 290, 317, 320, 321: Please use abbreviation for the morphine – MORPH or do not introduce the abbreviation at all. Please check the whole manuscript.

o   Line 239, 241: Please introduce once the abbreviation of square wave voltammetry (SWV) not twice.

o   Line 267, 268: Please give da abbreviation of limit of detection and limit of quantification: “Limit of detection (LOD) and limit of quantification (LOQ) were calculated…”

o   Line 273: Please add full name for abbreviation “RSD” when you first mentioned.

o   Line 316: Please use abbreviation for the carbon paste electrode – CPE.

Thank you for these comments and details. We accept all the suggestions and revised paper accordingly.

Reviewer 2 Report

The experimental paper entitled "Electrochemical determination of pain reliever morphine in biological matrices by tailoring FeWO4/CPE sensor" is in the scope of the journal, well readable, and quite actual. Therefore, the manuscript can be published in Biosensors after improvement and correction. 

The main comments are listed as follows:

1.         In the introduction section it should be mentioned the possible average/maximal opiates' level in human urine, and how it correlates with the blood level.

2.         Fig. 6(A) is represented very poorly. All the sensor’s curves should be normalized vs a baseline curve. In the current presentation, there is no observed any sensor output for 2 or 5 µM morphine vs baseline taken as zero. The outputs toward 20 µM and 35 µM morphine probably are marked wrong, as a signal for the smaller concentration is higher in comparison to the higher concentration of the target analyte.

3.         The caption of Fig. 6 is incorrect, as Fig. 6(A) represents sensor’s SWV voltammograms toward the addition of increasing concentrations of morphine, while Fig. 6(B) - calibration curve plotted using extracted data from the corresponding SWVs. It seems the authors copied the capture from figure 5 due to a mistake during the paper compilation.

4.         Is the sensor selective only to morphine, or can give the same output to different opiates? Have you analyzed its selectivity toward heroin; codeine; thebaine; hydrocodone; oripavine; or oxymorphone? It seems that the sensor should be nonselective to the listed opiates according to Scheme 1 of the paper. It should be discussed.

5.         A short discussion section focused on a comparison of the main characteristics of the proposed biosensor (LOD, linearity, sensitivity, selectivity, and stability) with the same type of known analogs is required.

6.         I would propose for authors to talk about the sensor's practical application very carefully, as a significant interfering impact from analyzed citric acid and dopamine is clearly observed even when they were tested in very low concentrations (it is not clear from the text, but I assume it was used 0.1 mM, this also should be clarified). Keeping in mind that the normal level of citric acid in urine could be elevated up to 12 mM and its interfering impact combined with the interference of other easily electrooxidising urine compounds like succinate, ascorbate, etc can completely mask the sensor output to the target analyte (opiates).

7.         The dilution step of urine used for spiking with morphine should be indicated, and the procedure of urine spiking clearly described. At the current representation, it is unclear if at this dilution level the real sample initially containing morphine could be estimated, taking into account the linearity of the sensor (up to 85 µM).

Author Response

Reviewer 2 comments:

The experimental paper entitled "Electrochemical determination of pain reliever morphine in biological matrices by tailoring FeWO4/CPE sensor" is in the scope of the journal, well readable, and quite actual. Therefore, the manuscript can be published in Biosensors after improvement and correction. 

Thank you for your efforts, positive comments and valuable suggestions.  

The main comments are listed as follows:

  1. In the introduction section it should be mentioned the possible average/maximal opiates' level in human urine, and how it correlates with the blood level.

Thank you for the suggestion. In the revised version of the manuscript we added these values in the section “Selectivity of the method and application in real-world samples analysis” with corresponding discussion and appropriate references. 

  1. Fig. 6(A) is represented very poorly. All the sensor’s curves should be normalized vsa baseline curve. In the current presentation, there is no observed any sensor output for 2 or 5 µM morphine vs baseline taken as zero. The outputs toward 20 µM and 35 µM morphine probably are marked wrong, as a signal for the smaller concentration is higher in comparison to the higher concentration of the target analyte.

Thank you for the suggestion. We revised this figure and, as well, after the reviewer suggestion we revised our linear range. In the revised version, new Figure 6 is provided and new discussion about linear range.

  1. The caption of Fig. 6 is incorrect, as Fig. 6(A) represents sensor’s SWV voltammograms toward the addition of increasing concentrations of morphine, while Fig. 6(B) - calibration curve plotted using extracted data from the corresponding SWVs. It seems the authors copied the capture from figure 5 due to a mistake during the paper compilation.

Thank you for noticing this. You have right, this was technical mistake. In the revised version appropriate caption for Figure 6. is provided.

  1. Is the sensor selective only to morphine, or can give the same output to different opiates? Have you analyzed its selectivity toward heroin; codeine; thebaine; hydrocodone; oripavine; or oxymorphone? It seems that the sensor should be nonselective to the listed opiates according to Scheme 1 of the paper. It should be discussed.

Thank you for this comment. In the revised version we provided additional discussion regarding to the method selectivity, however, due to absence of the standards we are not able to provide their electrochemical behavior studies. Appropriate references are provided.

  1. A short discussion section focused on a comparison of the main characteristics of the proposed biosensor (LOD, linearity, sensitivity, selectivity, and stability) with the same type of known analogs is required.

Thank you for your comment. In the revised version additional discussion focused on a comparison of the main characteristics of the proposed biosensor with the same type of known analogs is provided as Table 1. 

  1. I would propose for authors to talk about the sensor's practical application very carefully, as a significant interfering impact from analyzed citric acid and dopamine is clearly observed even when they were tested in very low concentrations (it is not clear from the text, but I assume it was used 0.1 mM, this also should be clarified). Keeping in mind that the normal level of citric acid in urine could be elevated up to 12 mM and its interfering impact combined with the interference of other easily electrooxidising urine compounds like succinate, ascorbate, etc can completely mask the sensor output to the target analyte (opiates).

Thank you for your suggestion. In the revised version we provided MORPH concentration during selectivity studies. We agree that concentration of the citric acid is much higher in the urine samples than our analyte, however, our practical applicability studies, additionally confirmed by HPLC method, were in excellent agreement with prepared samples. Probably some deeper studies need to be done to evaluate this phenomenon.

  1. The dilution step of urine used for spiking with morphine should be indicated, and the procedure of urine spiking clearly described. At the current representation, it is unclear if at this dilution level the real sample initially containing morphine could be estimated, taking into account the linearity of the sensor (up to 85 µM).

Thank you for the suggestions. This is not clearly explained in the original manuscript. The dilution was done 3 times. This is now added in the revised version and highlighted. Also, levels of morphine in urine samples was provided with appropriate references. As the found level of morphine in urine is up to 26 µM, sample dilution of 3 times is in the range of calibration curve. This is added in the revised version. 

Reviewer 3 Report

The manuscript titled “Electrochemical determination of pain reliever morphine in biological matrices by tailoring FeWO4/CPE sensor” by Miloš Ognjanović et al., deals with development of an electrochemical sensor for detection of Morphine from a biological medium at submicromolar concentration. Authors constructed a FeWO4 electrode with different Fe/W ratio and tested their electrochemical and morphological properties to fabricate a potent carbon paste electrode for morphine specificity. The method somewhat demonstrated a good selectivity and low detection limit for morphine with a wide dynamic range. The manuscript fits to the scope of the Biosensors and written well in general. However, prior publication the following concerns needs to be addressed:   

Major:

In section 3.5. “ Selectivity of the method and application in real-world samples analysis” authors performed experiments using urine samples from morphine negative volunteers. To report unbiased results, the experiment should be also performed on the samples using urine samples with morphine positive volunteers. Thus, the applicability of the developed method is highly questionable without the appropriate controls.

Table 1. Should also include data/results from other methods to compare the new electrochemical method to determine morphine, for example HPLC analysis.   

Minor:

Abstract, the first sentence: “Morphine (MORPH) is mainly used as a pain reliever and is also often used recreationally or to make other illicit opioids”. The use of morphine is highly regulated, and often prohibited to use recreationally, as many other opioid-family substances. These are highly addictive compounds, hence the phrase “and is also often used recreationally” must be removed or appropriate justification must be added.

The manuscript specifies that method applied to urine samples as a type of biological fluid. The manuscript’s title should be more specific that this was specifically for urine, unless other biological matrices are tested. 

P.2 lanes 52-55: paragraph is missing citations to the known detection and quantification of morphine.

P.2 lanes 79-80: The source of MORPH is not provided, was it purchased or synthesized, pure substance of mixed? 

Author Response

Reviewer 3 comments:

The manuscript titled “Electrochemical determination of pain reliever morphine in biological matrices by tailoring FeWO4/CPE sensor” by Miloš Ognjanović et al., deals with development of an electrochemical sensor for detection of Morphine from a biological medium at submicromolar concentration. Authors constructed a FeWO4 electrode with different Fe/W ratio and tested their electrochemical and morphological properties to fabricate a potent carbon paste electrode for morphine specificity. The method somewhat demonstrated a good selectivity and low detection limit for morphine with a wide dynamic range. The manuscript fits to the scope of the Biosensors and written well in general. However, prior publication the following concerns needs to be addressed:   

Thank you for the positive comments and your efforts which will strongly improve our manuscript.

Major:

In section 3.5. “ Selectivity of the method and application in real-world samples analysis” authors performed experiments using urine samples from morphine negative volunteers. To report unbiased results, the experiment should be also performed on the samples using urine samples with morphine positive volunteers. Thus, the applicability of the developed method is highly questionable without the appropriate controls.

Thank you for your comments and suggestions. We provided samples from healthy volunteers because we have approval to work with this type of samples. Obtaining samples from morphine-positive volunteers is currently impossible, due to ethical issues and extremely complicated procedures for working with them. Based on that, we used the addition method to examine the practical applicability of the method and its accuracy and precision.

Table 1. Should also include data/results from other methods to compare the new electrochemical method to determine morphine, for example HPLC analysis.   

Minor:

Abstract, the first sentence: “Morphine (MORPH) is mainly used as a pain reliever and is also often used recreationally or to make other illicit opioids”. The use of morphine is highly regulated, and often prohibited to use recreationally, as many other opioid-family substances. These are highly addictive compounds, hence the phrase “and is also often used recreationally” must be removed or appropriate justification must be added.

Thank you for your suggestion. The phrase “and is also often used recreationally” is removed from the text.

The manuscript specifies that method applied to urine samples as a type of biological fluid. The manuscript’s title should be more specific that this was specifically for urine, unless other biological matrices are tested. 

Thank you for the suggestion. We revised manuscript title accordingly.

P.2 lanes 52-55: paragraph is missing citations to the known detection and quantification of morphine.

Thank you for the suggestion. Missing citations are provided.

P.2 lanes 79-80: The source of MORPH is not provided, was it purchased or synthesized, pure substance of mixed? 

Thank you for noticing this. This information is provided in the text.

Round 2

Reviewer 2 Report

It is pleasant to see that the authors have accepted my recommendations and given adequate responses to some tricky questions.
The paper could be accepted for publication in Biosensors in its current (revised) form.

Author Response

We are very grateful for the positive comments.

Reviewer 3 Report

One of my major concerts has not been addressed. 

Additional experiment using qualitative analysis (chromatography methods for example) of Morphine concentration is required as a comparison to the developed tool.

Author Response

Thank you for the suggestions. In the revised version we provided additional Figure  8. with HPLC chromatograms for morphine standard solutions and tested samples at two different wavelengths.

Figure 8. HPLC chromatograms obtained for morphine standard solutions and tested samples.
